# Inverse Solution Error Analysis and Correction of Beam Steering System Based on Risley Prisms

**Yinuo Song** [1,2], **Shijie Gao** [1,2,*], **Jiabin Wu** [1], **Shuaifei Wang** [1,2] **and Li Huo** [1,2]

1   Changchun Institute of Optics, Fine Mechanics and Physics, Chinese Academy of Sciences, Changchun 130033, China; songyinuo17@mails.ucas.ac.cn (Y.S.); wujb@ciomp.ac.cn (J.W.); wangshuaifei20@mails.ucas.ac.cn (S.W.); huoli18@mails.ucas.ac.cn (L.H.)
2   University of Chinese Academy of Sciences, Beijing 100049, China
*   Correspondence: gaoshijie@ciomp.ac.cn

**Abstract:** The pointing accuracy of the Risley prisms beam-steering system mainly depends on the accuracy of the solution method and the impact of the error on the solution. To improve pointing accuracy, the impact of systematic errors on the inverse solution precision is investigated and a correction method is proposed. First, a more accurate error model for Risley prisms is established, and the errors are obtained by the parameter identification method, which corrects the forward solution error. Second, we explain the reason for the error generated by the inverse solution and analyze the variation of the inverse solution error magnitude with the beam deflection angle. A correction method based on pointing-field transformation is proposed. Finally, simulations and experiments are performed to verify the feasibility of the method. Experimental results show that when the beam deflection angle is equal to $0.1°$, the maximum and RMS values of the pointing error are reduced by 94.08% and 95.18%, respectively.

**Keywords:** Risley prisms; pointing error; laser communication; inverse solution; error correction

## 1. Introduction

Risley prisms are a widely used beam-steering mechanisms, with their compact structure, high precision, low moment of inertia, and low cost [1,2], they are often used as beam-pointing devices and beam-scanning devices in laser communication [3], large-FOV imaging [4], photoelectric tracking [5], lidar [6], laser scanning [7], and other fields. Risley prisms consist of a pair of optical wedges that rotate around a common axis independently, which enables high-resolution arbitrary beam-pointing over a wide range.

The features and advantages of Risley prisms make them suitable as a coarse or fine pointing mechanism for laser communication acquisition, tracking, and pointing (ATP) systems, especially for inter-satellite and airborne laser communication. Figure 1 shows the optical layout of the laser communication system based on the Risley prism's coarse tracking mechanism. The accuracy of the beam-steering mechanism of the ATP system is one of the key factors affecting the system's performance. The pointing accuracy of the Risley prism is mainly determined by two factors: the accuracy of the solution method and the impact of the error on the solution. The former refers to the precision of solving the mathematical relationship between the rotation angle of prisms $\theta_1, \theta_2$ and the spatial orientation of the outgoing beam $\Phi, \Theta$. The definition of the prism rotation angle $\theta_1, \theta_2$ and the deflection angle and azimuth of the outgoing beam $\Phi, \Theta$ are shown in Figure 2. The trapezoid formed by the dotted line in the figure is the principal section of the prism, and the red arrows represent the light rays.

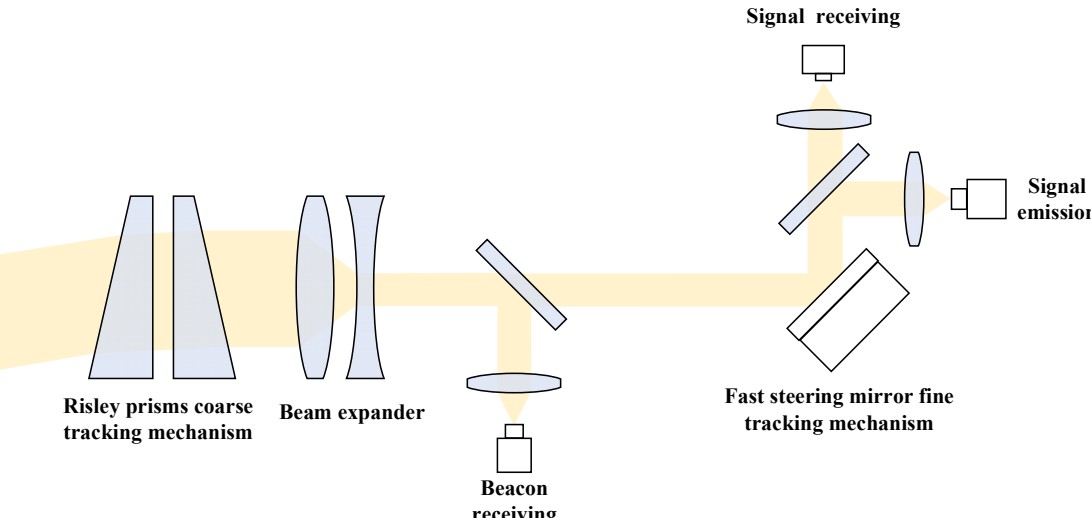

**Figure 1.** Optical layout of a laser communication system with a Risley prism coarse tracking mechanism.

The solution consists of two aspects: solving the direction of the emitted light according to the prism's rotation angles, called the forward solution, and solving the prism's rotation angles with a known direction of the emitted beam, called the inverse solution. Yang used the vector form of Snell's law to perform nonparaxial ray tracing and gave an exact forward analytical solution for the first time [8]. For the inverse solution, the two-step method [9], table look-up method, third-order approximate expression solution [10], damped least-squares iterative solution [11], and forward iterative optimization solution [12] have been proposed.

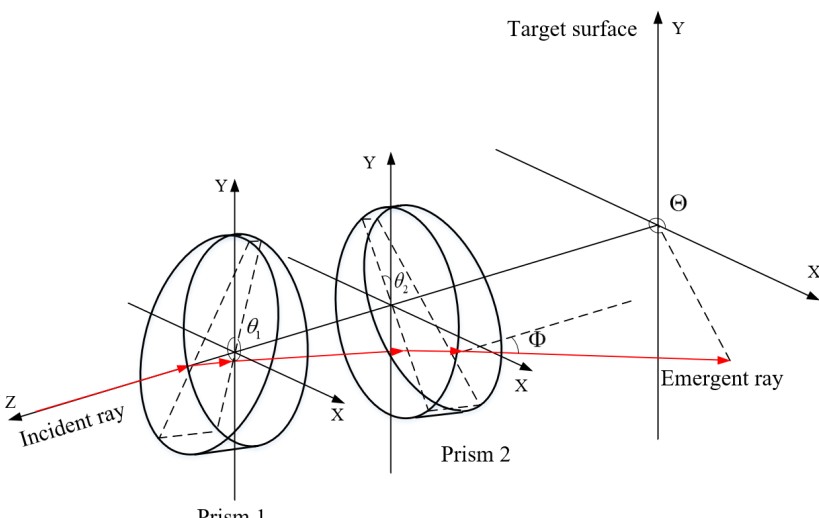

**Figure 2.** Schematic diagram illustrating the coordinate systems for Risley prisms. The incident ray, rotation axis of prisms, and Z-axis are collinear.

The systematic errors of Risley prisms also impact the accuracy of the forward and inverse solution. The systematic error refers to the deviations between the actual and theoretical parameters of the Risley prisms system, which usually has a greater impact than the solution errors. For the impact of systematic errors on the performance and forward solution accuracy, a lot of work has been carried out. Horng investigated the optical distortions of scan patterns caused by the component errors and alignment errors of prisms [13]. Zhou et al. analyzed the influence of component errors, prism orientation errors, and assembly errors on pointing accuracy. The allowances of the error sources for a given pointing accuracy were evaluated [14].

There are relatively few studies focusing on compensating for the impacts of system errors. Bravo-Medina et al. proposed an error correction method based on the paraxial approximation model [15]. Under the paraxial approximation model, an error vector was added to compensate for the system error. The method model is simple, easy to implement, and considers the influence of the thickness and spacing of the prism on the outgoing beam. However, the solutions obtained by the paraxial approximation method will deviate from the exact solution, especially when the deflection angle is large [16]. So, it is not suitable for large-angle deflection applications.

Li et al. proposed an error parameter identification method [17]. This method is based on an analytical model of achromatic Risley prisms. The genetic algorithm was used to fit the actual parameters of the physical model, which can achieve higher accuracy than the approximate model. However, not all errors are included in the error fitting of the article. The Risley prism used in the article has a small pointing range ($\pm 3°$), so errors that have little effect on pointing accuracy are ignored. In a system with a large pointing range, the beam-pointing deviation caused by these errors cannot be ignored.

However, the above studies all focus on the impact of errors on the forward solution; there are few studies on the impact of errors on the inverse solution. Considering the accuracy and the real-time and data volume requirements of laser communication, the two-step method is the most suitable inverse solution method. Since the solution of the two-step method is based on the ideal motion trajectory of the spot, the offset of the actual motion trajectory caused by the systemic error of the prism will cause the pointing error of the inverse solution. This paper investigates the inverse solution error of Risley prisms induced by systemic error and proposes a method based on pointing-field transformation for correction.

The outline of this paper follows. A more accurate error model is established in Section 2. In Section 3, the causes of the inverse solution error are explained, and simulations are performed to analyze the impact of the inverse solution error on pointing accuracy. Furthermore, the correction algorithm is proposed. In Section 4, the feasibility of the correction algorithm is verified by simulation and experiment. Conclusions are drawn in Section 5.

## 2. Error Model of Risley Prisms and Forward Solution Error Correction Method

### 2.1. Forward Solution and Inverse Solution of Risley Prisms

The precise forward and inverse solution of Risley prisms is derived by nonparaxial ray tracing through the vector form of Snell's law shown in Equation (1) [18].

$$\vec{A_r} = \frac{n_i}{n_r}\left[\vec{A_i} - \left(\vec{A_i} \cdot \vec{N}\right) \cdot \vec{N}\right] - \sqrt{1 - \frac{n_i^2}{n_r^2} + \frac{n_i^2}{n_r^2}\left(\vec{A_i} \cdot \vec{N}\right)^2} \cdot \vec{N} \tag{1}$$

where $n_i$ and $n_r$ are the refractive indices of the medium on both sides of the refractive surface, $\vec{N}$ is the unit normal vector of the refraction interface, $\vec{A_i}$ is the unit vector of the incident light, and $\vec{A_r}$ is the vector of the exit light. The notations of the prism surface normal vector and the ray direction vector are shown in Figure 3. The refractive indices of prism 1 and prism 2 are denoted by $n_1$ and $n_2$, respectively. The direction of the incident light is defined as the reverse direction of the Z-axis, so its vector is $\vec{A_1} = (0, 0, -1)$.

Calculations are performed using Equation (1) in each refractive surface of the Risley prism, and the direction vector of the outgoing beam can be obtained:

$$\vec{A_5} = (K, L, M) \tag{2}$$

where $K, L, M$ are the final calculation results; they are functions of the prism rotation angles $\theta_1$ and $\theta_2$, and they are also the components of the outgoing light vector on the $X$, $Y$, and $Z$ axis.

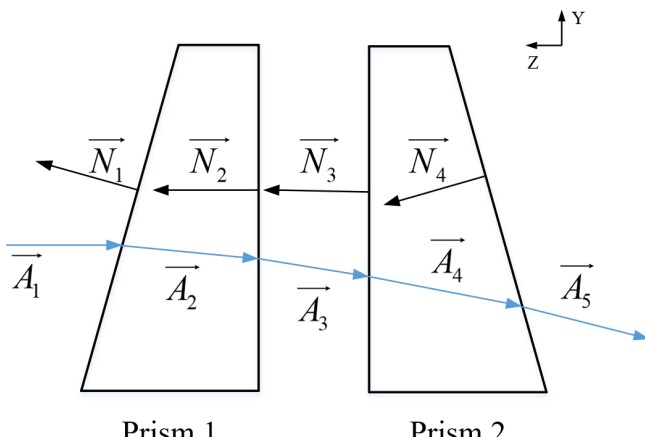

**Figure 3.** Diagrams illustrating the notation of the normal vector for each prism and the incident ray vector. The black arrows represent the surface normal vectors, and the blue arrows represent the incident and refracted light vectors.

The deflection angle and azimuth of the outgoing beam can be calculated through the direction vector:

$$\Phi = \arccos(-M) \tag{3}$$

$$\Theta = \begin{cases} \arctan\left(\frac{L}{K}\right), & K \geq 0 \text{ and } L \geq 0 \\ \arctan\left(\frac{L}{K}\right) + 2\pi, & K \geq 0 \text{ and } L < 0 \\ \arctan\left(\frac{L}{K}\right) + \pi, & K < 0 \end{cases} \tag{4}$$

The inverse solution of Risley prisms is solved by the two-step method. Since the angle between the two prisms determines the deflection angle of the outgoing beam. The first step is to calculate the included angle between two prisms $\Delta\theta = |\theta_1 - \theta_2|$, which corresponds to the deflection angle $\Phi_t$ of the desired beam-pointing position. The current prism angle $\Delta\theta'$ is known, assuming that one prism remains stationary and the other one rotates $|\Delta\theta - \Delta\theta'|$, calculating the azimuth angle $\Theta_0$ of the beam at this time. This step makes the deflection angle of the outgoing beam reach the target value.

When two prisms rotate 360° simultaneously with the same speed and direction at a certain included angle, the deflection angle of the outgoing beam remains unchanged, and the azimuth angle changes continuously. Hence, the trajectory of the light on the OXY surface is a circle centered on the intersection of the Z-axis and the OXY surface. We use the abbreviation "TLS" to specifically refer to this trajectory. Therefore, the second step is to rotate the two prisms by $\Delta\Theta = \Theta_t - \Theta_0$ simultaneously in the same direction. According to the two-step method process, the final rotation angles required of prisms can be obtained:

$$\begin{cases} \theta'_1 = \Theta_t - \Theta_0 + |\Delta\theta - \Delta\theta'| \\ \theta'_2 = \Theta_t - \Theta_0 \end{cases} \tag{5}$$

or

$$\begin{cases} \theta'_1 = \Theta_t - \Theta_0 \\ \theta'_2 = \Theta_t - \Theta_0 - |\Delta\theta - \Delta\theta'| \end{cases} \tag{6}$$

Since there are two ways of rotation in the first step, two sets of solutions can be obtained.

### 2.2. Error Model of Risley Prisms

The error model of the Risley prisms is obtained by introducing systemic errors into the mathematical model of the Risley prisms. The systemic errors of Risley prisms can be divided into component errors and assembly errors [14]. Component errors refer to the deviation of the prism wedge angle and the refractive index from the design value, which may change with temperature or wavelength. Assembly errors include prism rotation

axis tilt, prism tilt, incident light tilt, and prism rotation angle error. Their description and notation are shown in Figure 4.

Take prism 1 as an example. Ideally, the prism rotation axis coincides with the $Z$ axis, $Z_1$ is the actual rotation axis which deviates from the $Z$ axis, and the error between the two is represented by $H_{R1}, V_{R1}$. $V_{R1}$ is the angle between the $Z_1$ axis and plane OZX, which represents the vertical error of the $Z_1$ axis. $H_{R1}$ is the angle between the projection of the $Z_1$ axis on the plane OZX and the $Z$ axis, and represents the horizontal error of the $Z_1$ axis. The coordinate system $X_2Y_2Z_2$ denotes the coordinate system of the tilting prism; the tilt error of the prism in the vertical and horizontal directions are denoted by $H_{P1}, V_{P1}$, respectively.

In the same way, the tilt errors of the incident light in the vertical and horizontal directions are represented by $H_I, V_I$, respectively.

In Figure 2, the configuration when the rotation angle of the prisms is $\theta_1 = \theta_2 = 0$ is defined as the zero position of the Risley prisms. The analytical model of the Risley prisms is derived when the prisms are in the zero position; therefore, the rotation angle of the prisms is calculated from the zero position. In practice, the prisms cannot be precisely mounted at the zero position, as the prism rotation angle error refers to the angle between the actual initial mounting position of the prism and the zero position. The symbols $\delta\theta_1$ and $\delta\theta_2$ are used to denote the rotation angle errors of prism 1 and prism 2, respectively.

According to the errors defined in Figure 4, the incident light vector $\vec{A_1}$, the normal vectors of the prisms' four surfaces $\vec{n}_{110}, \vec{n}_{120}, \vec{n}_{210}, \vec{n}_{220}$, and the prism rotation axis vectors $\vec{k}_1, \vec{k}_2$ can be obtained. Let $\vec{k}_1 = (k_x, k_y, k_z)$; the transformation matrix of any vector rotating around $\vec{k}_1$ can be obtained by using Rodrigues' rotation formula:

$$R_1 = E\cos\theta_1 + (1-\cos\theta_1)\begin{pmatrix} k_x \\ k_y \\ k_z \end{pmatrix}\begin{pmatrix} k_x & k_y & k_z \end{pmatrix} + \sin\theta_1 \begin{pmatrix} 0 & -k_z & k_y \\ k_z & 0 & -k_x \\ -k_y & k_x & 0 \end{pmatrix} \quad (7)$$

where $E$ is the unit matrix, and $\theta_1$ is the rotation angle of prism 1.

When prism 1 rotates around $\vec{k}_1$, the normal vectors of its two surfaces are $\vec{n}_{11} = R_1 \cdot \vec{n}_{110}$ and $\vec{n}_{12} = R_1 \cdot \vec{n}_{120}$. In the same way, the normal vectors of the two surfaces of prism 2 can be obtained: $\vec{n}_{21} = R_2 \cdot \vec{n}_{210}$ and $\vec{n}_{22} = R_2 \cdot \vec{n}_{220}$, where $R_2$ the transformation matrix of any vector rotating around $\vec{k}_2$.

Substituting the above vectors into Formula (1) in turn for calculation, the emitted light vector $\vec{A_5} = (K', L', M')$ can be derived.

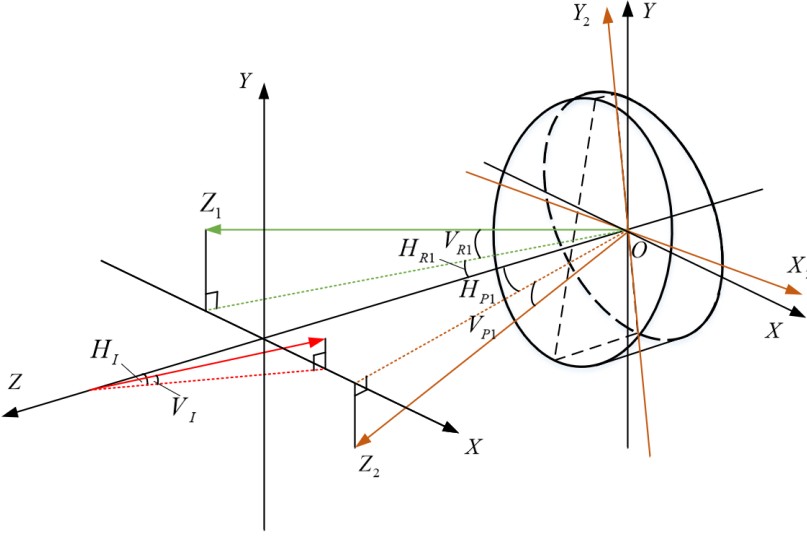

**Figure 4.** Diagram illustrating the assembly errors of the prisms and their notation.

### 2.3. Error Parameter Identification

On the basis of the error model, the exact or equivalent values of errors can be obtained by parameter identification [17]. As shown in Equation (8), the error model describes the relationship between prism rotation angles $\theta_1, \theta_2$ and errors and the deflection angle and azimuth of the outgoing beam $\Phi, \Theta$.

$$f(\theta_1, \theta_2, \delta\theta_1, \delta\theta_2, H_I, V_I, H_P, V_P, H_R, V_R) = (\Phi, \Theta) \tag{8}$$

The essence of the parameter identification method is to fit the exact or equivalent value of the error through the experimental data.

Among the several systemic errors in the error model, the incident light tilts $H_I$, $V_I$ and prism rotation angle errors $\delta\theta_1$, $\delta\theta_2$ are constants, while the bearing rotation axis wobble is a systematic error that changes periodically with the rotation of the bearing. Therefore, we set the prism rotation axis tilt $H_R$, $V_R$ as a function of the prism rotation angles $\theta_1$ and $\theta_2$. After the prism is mounted on the bearing, the angle between the prism and its rotation axis is a fixed value, so we use the rotation axis tilt to represent the prism tilt $H_P$, $V_P$. With sufficient data and a suitable optimization algorithm, the exact or equivalent value of the errors shown in Equation (8) can be fitted by the least-squares method.

## 3. Causes and Correction of Inverse Solution Error

### 3.1. Causes of Inverse Solution Error

Using the two-step method for inverse solution in the error model faces two problems. First, the introduction of errors complicates the function relationship between outgoing beam and prism rotation angle. Moreover, we set the prism rotation axis tilt as a function of the rotation angle, which makes the analytical solution unobtainable. This problem can be solved by a fast numerical solution through dichotomy [17].

Second, the incident light tilt and the prism rotation axis tilt will cause the pointing field to deviate from the theoretical pointing field and the TLS distortion. They will cause pointing errors in the second step.

The detailed causes of the inverse-solution error are discussed next. The simulated and experimentally measured TLS is shown in Figure 5. The blue curve is the ideal trajectory and the black curve is the trajectory with error impacts. We simulate with the actual experimental setup and system parameters. The outgoing beam is received by a camera and the TLS is the trajectory of the beam on the camera sensor. The system parameters used in the simulation are shown in Table 1.

**Table 1.** System parameters used in simulation.

| Parameter | Value |
| --- | --- |
| Refractive Index | 1.94595 |
| Wedge Angle | 17.2° |
| Lens Focal Length | 81.09 mm |
| Pixel Size | 5.3 μm |

Figure 5a depicts the effect of the incident light tilt on the TLS, where the incident light tilt deflects the pointing field. Figure 5b depicts the TLS when the prism is tilted; it can be seen that the prism tilt changes the size of the pointing field. Figure 5c,d depict the impacts on the TLS when the prism axis tilt error is a constant and a function of the prism rotation angle, respectively. The prism axis tilt can both deviate the beam-pointing field and deform the TLS. Figure 5e shows the actual TLS obtained by experiment, which contains the effects of all errors.

Figure 6 illustrates the generation principle of the inverse solution error. The ideal pointing field is a circular region that consists of an infinite number of concentric circles centered on $O$. Each concentric circle corresponds to an included angle of prisms $\Delta\theta$ and a deflection angle $\Phi$ of the incident light. The tilt of the incident light deviates the pointing

field into a circular region centered on $O'$. Assuming that point A is the initial position of the light and point C is the target pointing position, the circles where they are located correspond to the beam deflection angles $\Phi_1$ and $\Phi_2$. According to the two-step method solution process, $\Delta\theta$ is first obtained, and then one prism rotates $|\Delta\theta - \Delta\theta'|$ to achieve the desired beam deflection angle, at which time the beam motion trajectory is from point A to point B. The second step is to calculate the azimuth difference $\Delta\Theta$ between point B and point C. Then the two prisms rotate $\Delta\Theta$ in the same direction. Ideally, the beam will point from point B to point C. However, the actual trajectory of the light shifts from the blue circle to the red circle, so the actual trajectory of the light is from point B to point D. The deviation between the actual pointing position of beam D and the target position C is the inverse solution error.

The prism rotation axis tilt will not only cause the beam-pointing field deviation, but also makes the TLS no longer a circle (as shown in Figure 5c,d). In a word, the inverse solution error comes from the deflection and distortion of the TLS caused by incident light tilt and prism rotation axis tilt. They lead to the beam azimuth and deflection angle changing at the same time in the second step; thus, the decoupling of azimuth and deflection angle cannot be realized.

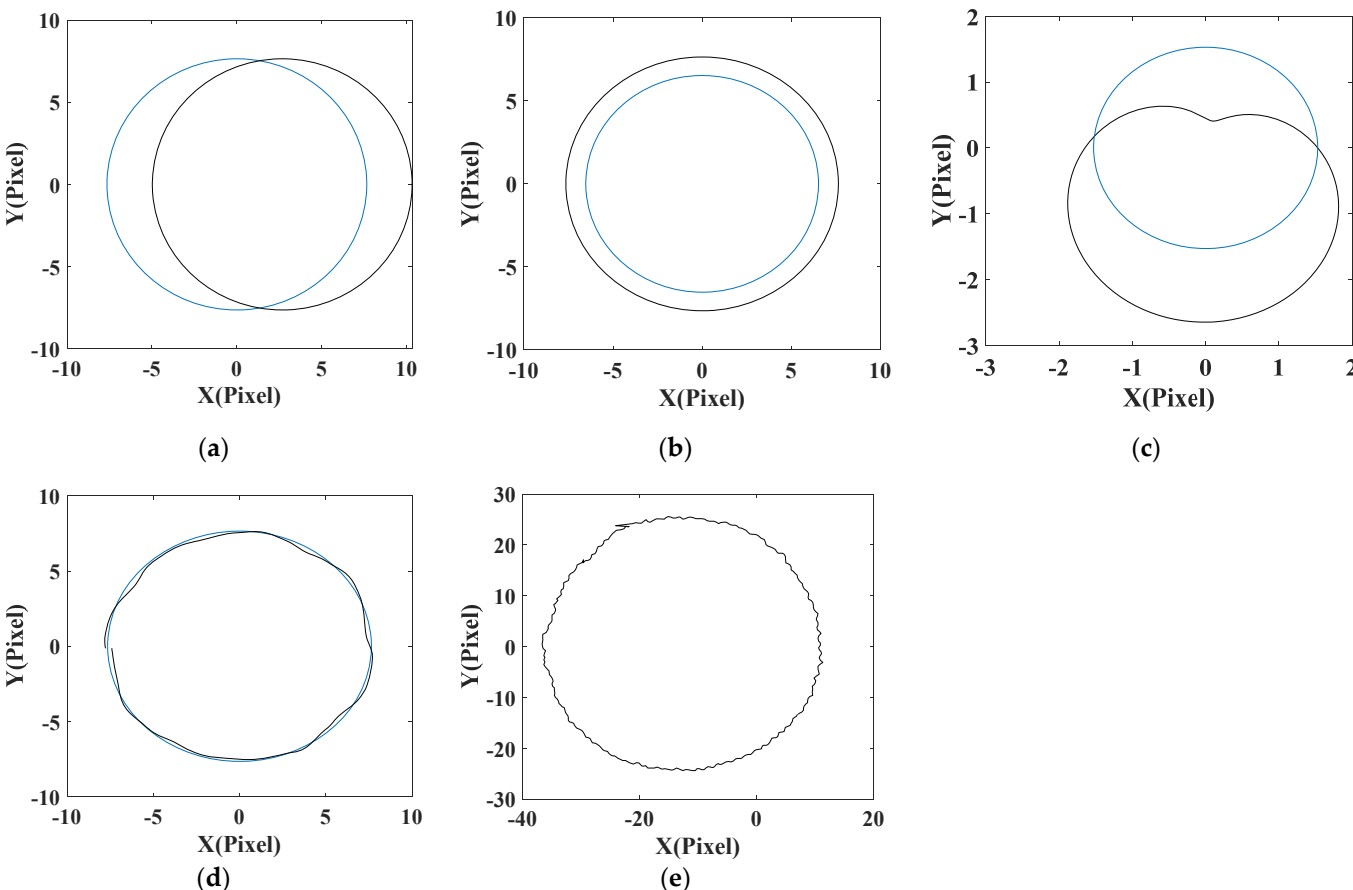

**Figure 5.** Impact of errors on the outgoing beam. The blue curve is the ideal TLS and the black curve is the TLS with error impacts. (**a**) Beam-pointing field deviation caused by the incident light tilt. (**b**) Beam-pointing field radius increase caused by prism tilt. (**c**,**d**) Beam-pointing field displacement and TLS distortion caused by the prism rotation axis tilt error; the prism rotation axis tilt is constant in (**c**), and this error is a function of the prism rotation angle in (**d**). (**e**) The actual TLS obtained in experiment.

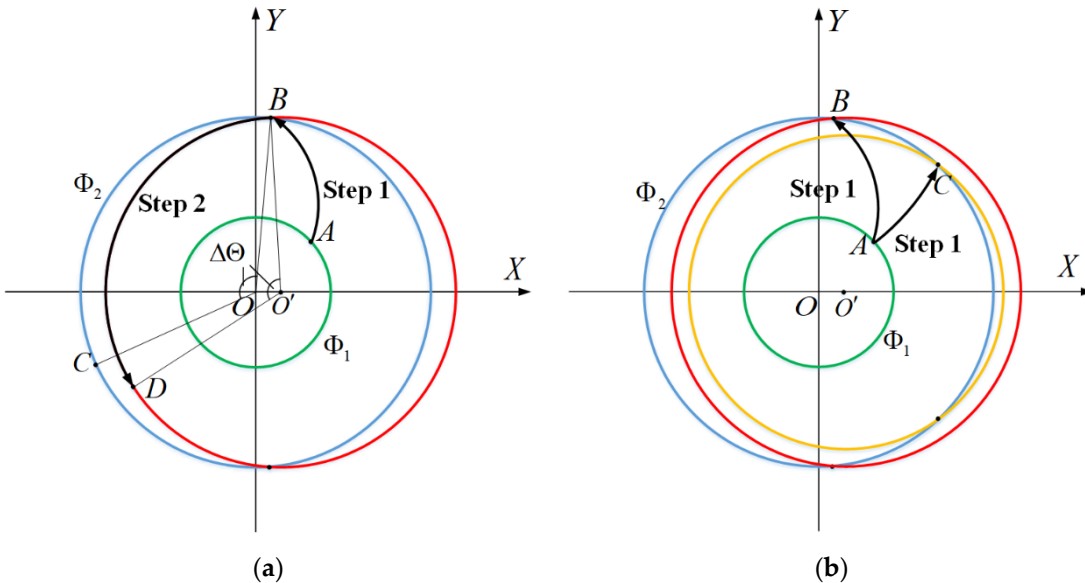

(a)                                                    (b)

**Figure 6.** (**a**) Generation principle of the inverse solution error. The green circle represents the circle corresponding to the deflection angle of the initial position in the ideal pointing field, and the blue circle represents the circle corresponding to the deflection angle of the target position. The red circle represents the circle corresponding to a solution in the actual pointing field. (**b**) The principle of generating multiple solutions in inverse solution. The meanings of the green and blue circles are the same as in Figure 6a. The red and yellow circles represent the circles corresponding to the two solutions in the actual pointing field, respectively.

The pointing field deviation and the TLS distortion also cause multiple solutions in the first step. As shown in Figure 6b, there are two ways of rotation in the first step, corresponding to the two prisms' included angles. The errors generated by these two ways in the second step are different. In fact, each TLS in the actual pointing field intersects with the blue circle corresponding to a solution.

When analyzing the inverse solution error in the entire pointing field, only the maximum error is taken; the curve of the inverse solution error versus $\Delta\Theta$ is shown in Figure 7. We separately analyze the variation trend of the pointing errors with $\Delta\Theta$ caused by $H_I$, $V_I$, $H_{R1}$, $V_{R1}$, $H_{R2}$, $V_{R2}$. The above errors are independent of each other, and the values of these errors are set to $0.1°$.

For the same error value, the inverse solution error caused by the incident light tilt is the largest. The error caused by the rotation axis tilt of prism 2 is much larger than that caused by the prism 1 rotation axis tilt, which is due to the fact that prism 2 will cause a greater pointing-field deviation than prism 1 under the same error. Table 2 summarizes the magnitude and causes of the inverse solution error generated by different parameters.

**Table 2.** The magnitude and causes of the inverse solution error generated by different parameters.

| Parameters | Magnitude of Error | Causes of Error |
|---|---|---|
| $H_I = 0.1°$ | $0.2002°\sim0.3233°$ | Pointing field deviation |
| $V_I = 0.1°$ | $0.2002°\sim0.3233°$ | Pointing field deviation |
| $H_{R1} = 0.1°$ | $0.0084°\sim0.0100°$ | Point field deviation & TLS deformation |
| $V_{R1} = 0.1°$ | $0.0042°\sim0.0051°$ | Point field deviation & TLS deformation |
| $H_{R2} = 0.1°$ | $0.0059°\sim0.1241°$ | Point field deviation & TLS deformation |
| $V_{R2} = 0.1°$ | $0.0063°\sim0.1255°$ | Point field deviation & TLS deformation |

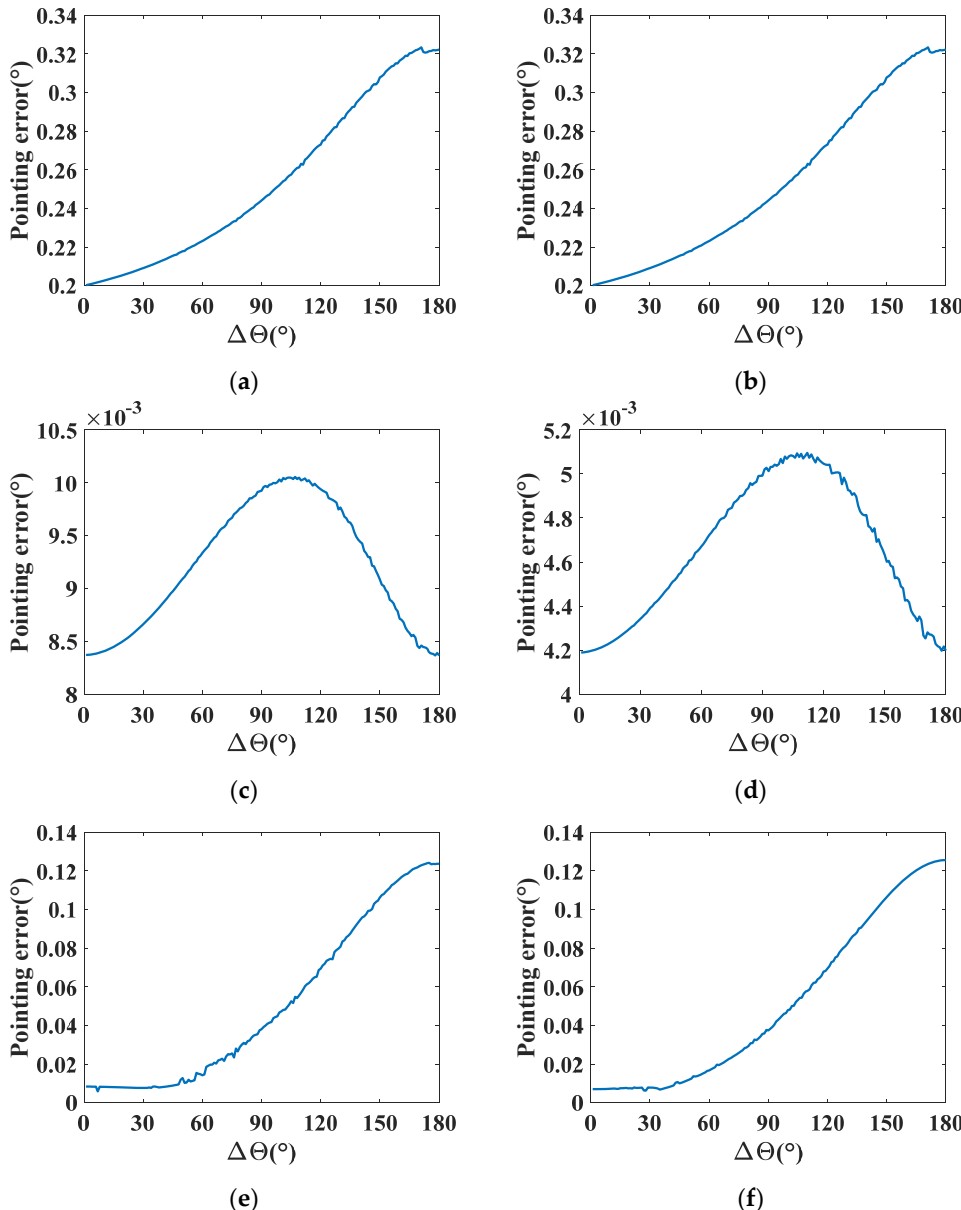

**Figure 7.** The inverse solution error caused by different systematic errors. (**a**) The incident light horizontally tilted by 0.1°. (**b**) The incident light vertically tilted by 0.1°. (**c**) The prism 1 rotation axis horizontally tilted by 0.1°. (**d**) The prism 1 rotation axis vertically tilted by 0.1°. (**e**) The prism 2 rotation axis horizontally tilted by 0.1°. (**f**) The prism 2 rotation axis vertically tilted by 0.1°.

### 3.2. Inverse Solution Error Correction Method

According to the above analysis, both the prism rotation axis tilt and the incident light tilt will cause the deviation of the pointing field. Moreover, the TLS deformation caused by the prism rotation axis tilt is much smaller than its displacement. So, the inverse solution error mainly comes from the deviation of the pointing field. If we transform the light spot coordinates to the actual pointing field coordinate system for a solution, the main inverse solution error caused by the pointing-field deviation can be eliminated.

However, as shown in Figure 8, the incident light tilt and the prism rotation axis tilt also cause the centers of the TLS to no longer coincide. This means that the points located on different TLS have different coordinate transformation values. It is necessary to first determine upon which TLS the target point is located. To solve this problem, a two-dimensional table needs to be established for table checking, but this would increase

the amount of data and computation significantly, which is not in line with our original intention of using the two-step method.

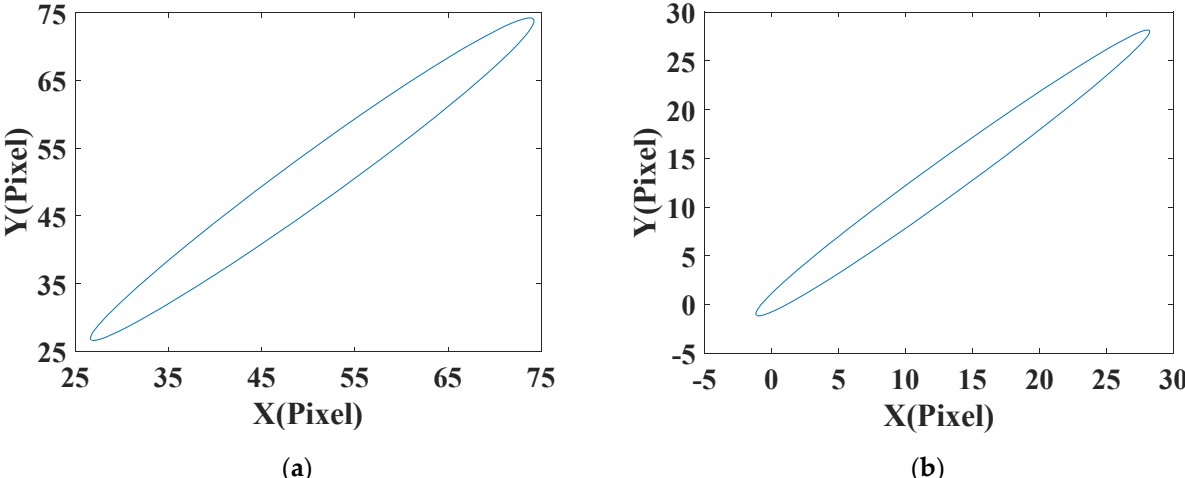

**Figure 8.** The center of the TLS when $\Delta\theta$ changes from $1°$ to $360°$. (**a**) $H_I = V_I = 0.1°$. (**b**) $H_{R2} = V_{R2} = 0.1°$.

For this problem, by the aid of the error model, we obtain the coordinate transformation value that minimizes the overall inverse solution error of all points corresponding to one target beam deflection angle $\Phi$ through the optimization method, and form a one-dimensional table. The procedures for the inverse solution are described below:

**Step 1.** Find the coordinate transformation value of target position by table look-up. Then transform the target position to the actual pointing-field coordinate system by coordinate transformation. Finally, calculate the deflection angle and azimuth angle $(\Phi', \Theta')$ of the target position in the new coordinate system.

**Step 2.** Let $H_I$, $V_I$, $H_R$, $V_R = 0$ in the error model to eliminate the deviation of the pointing field. Then use the two-step method to calculate the prism rotation angle corresponding to $(\Phi', \Theta')$ under the current model.

This method simultaneously calculates the coordinate transformation values of all points corresponding to a deflection angle, which can greatly reduce the amount of data and number of calculations.

## 4. Simulation and Experiment

### 4.1. Simulation

The parameters of the system used in the simulation experiment are shown in Table 1. The process of the simulation experiment is as follows:

1. Given the values or expressions $E_0$ of the systemic prism errors, substituting them into the error model to obtain a series of prism rotation angles $(\theta_1, \theta_2)$ and their corresponding outgoing beam positions $(K, L, M)_{error}$. The ideal beam-pointing position $(K, L, M)_{ideal}$ corresponding to $(\theta_1, \theta_2)$ is obtained through the ideal model.

2. Using the parameter identification method, the error values or expressions $E_1$ are fitted by the data obtained in process 1. Then $E_1$ is substituted into the error model to calculate the outgoing beam position $(K, L, M)_{correct}$ corresponding to the same rotation angle $(\theta_1, \theta_2)$. The forward solution errors before and after correction are calculated by $e_{forward1} = \arccos[(K, L, M)_{ideal} \cdot (K, L, M)_{error}]$ and $e_{forward2} = \arccos[(K, L, M)_{correct} \cdot (K, L, M)_{error}]$, respectively.

3. Given a series of target point positions $(K, L, M)_{target}$, the two-step method and the correction method are used to solve the corresponding prism rotation angles $(\theta_1, \theta_2)_{two-step\ method}$ and $(\theta_1, \theta_2)_{correct\ method}$.

4.   Substituting $(\theta_1, \theta_2)_{two-step\ method}$ and $(\theta_1, \theta_2)_{correction\ method}$ into the error model, the outgoing beam positions $(K, L, M)_{two-step\ method}$ and $(K, L, M)_{correction\ method}$ can be obtained. The inverse solution errors before and after correction are calculated by $e_{inverse1} = \arccos\left[(K, L, M)_{two-step\ method} \cdot (K, L, M)_{target}\right]$ and $e_{inverse2} = \arccos[(K, L, M)_{correction\ method} \cdot (K, L, M)_{target}]$, respectively.

The ranges of $\theta_1$ and $\theta_2$ are both 1° to 360°, and we generate a total of 129,600 pointing positions at 1° intervals, of which 144 positions are selected for fitting. Figure 9 shows the forward solution error before and after error fitting. Table 3 shows the maximum value, RMS value and descent rate of forward solution error before and after error fitting. It can be seen that the pointing error of the outgoing beam is significantly reduced after the error fitting. The maximum value and RMS of the beam-pointing error are reduced from 0.4099° and 0.2455° to 0.0064° and 0.0015°, respectively.

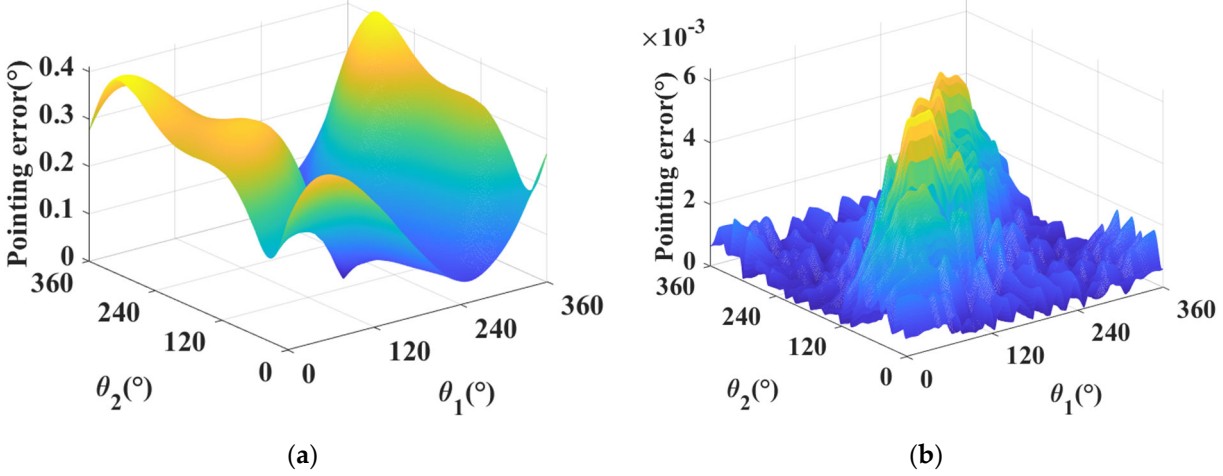

(**a**)                                                (**b**)

**Figure 9.** Simulation results of forward solution error correction. (**a**) Pointing error from forward solution before error fit. (**b**) Pointing error from forward solution after error fit.

**Table 3.** Simulation results of forward solution error correction.

|  | Pointing Error | |
|---|---|---|
|  | **Maximum** | **RMS** |
| **Before correction** | 0.4099° | 0.2455° |
| **After correction** | 0.0064° | 0.0015° |
| **Descent rate** | 98.44% | 99.34% |

Next, the simulation of the inverse solution error correction is performed. Figure 10a,b show the pointing error before and after correction in the whole pointing field, respectively. Figure 10c,d show the maximum error at each deflection angle, respectively. The maximum error drops from 0.4122° to 0.07873° and the RMS value of error drops from 0.2266° to 0.0291°. Table 4 shows the maximum value, RMS value and descent rate of inverse solution error before and after error correction. It can be seen that this method has a good effect on the pointing position correction of small deflection angles, but the residuals are larger for large deflection angles. The maximum error can meet the coarse tracking accuracy requirement of laser communication.

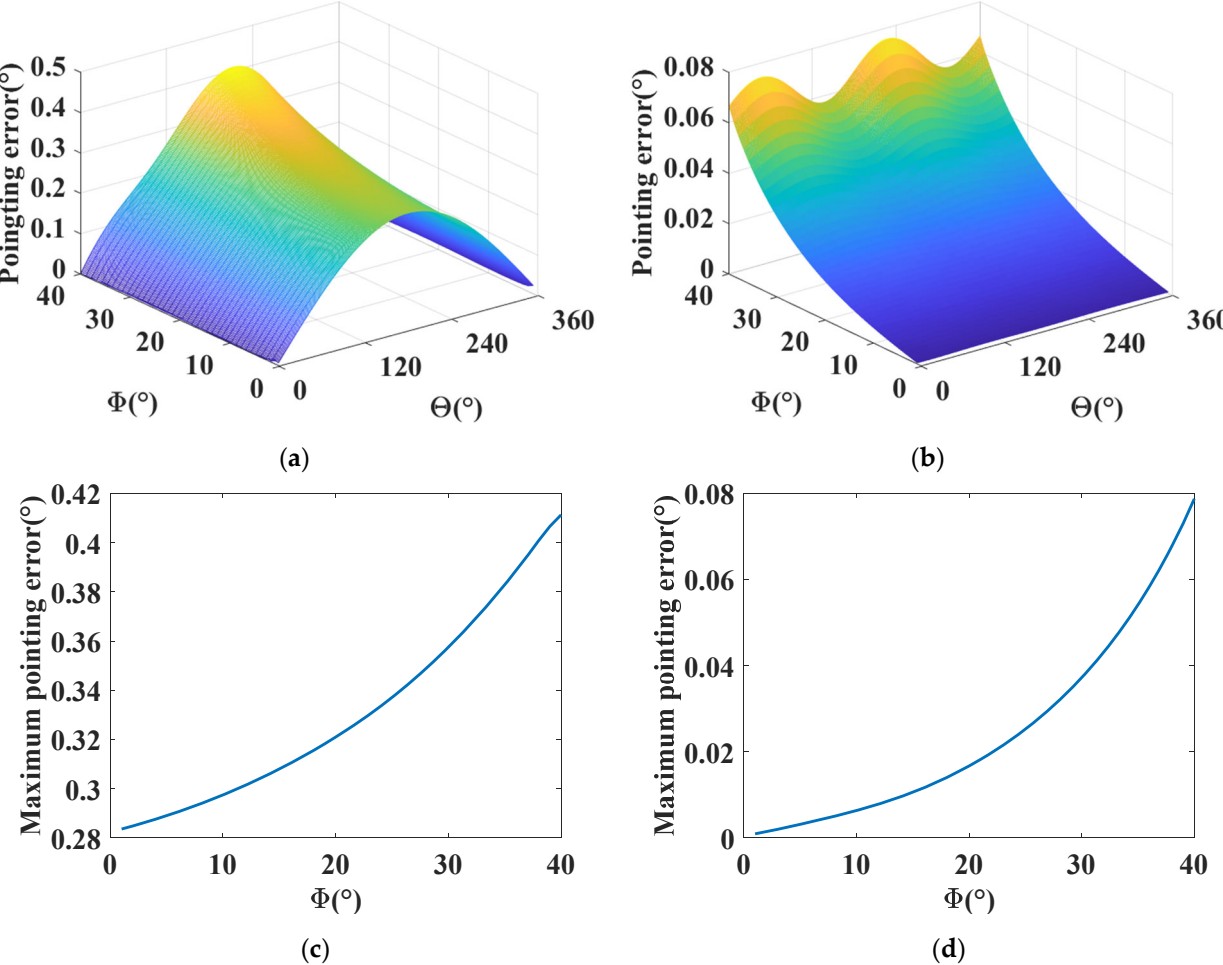

**Figure 10.** Simulation results of inverse solution error correction. (**a**) Inverse solution error in the whole pointing field before correction. (**b**) Inverse solution error in the whole pointing field after correction. (**c**) Maximum error in each deflection angle before correction. (**d**) Maximum error in each deflection angle after correction.

**Table 4.** Simulation results of inverse solution error correction.

|  | Pointing Error | |
|---|---|---|
|  | **Maximum** | **RMS** |
| **Before correction** | 0.4122° | 0.2266° |
| **After correction** | 0.07873° | 0.0291° |
| **Descent rate** | 80.9% | 87.16% |

*4.2. Experiment*

The experimental setup we used is shown in Figure 11. The laser is an MLL-III-640 produced by Changchun New Industry Optoelectronics Technology Co., Ltd.; the wavelength is $640 \pm 5$ nm, and the divergence angle is less than 1 mrad (full angle). The prism wedge angle is 17.2°, the refractive index is 1.94595, and the prism diameter is 48 mm. Each bearing of the prism is fitted with an encoder made by Renishaw to measure the angle of the rotation of the prism. The resolution of the encoder is 0.02″. The focal length of the camera lens is 81.09 mm (measured by the image magnification method); the camera made by Basler has $1024 \times 1024$ pixels, and the pixel size is 5.3 μm × 5.3 μm. The resolution of the camera after pixel subdivision is 1.348″, and the deflection angle of the light is calculated from the coordinates of the spot on the camera sensor obtained by the centroid extraction algorithm. The field of view of the camera is 3.8287°. Since the camera FOV is much smaller

than the pointing field, the experiments are only conducted in the angular range of the camera FOV.

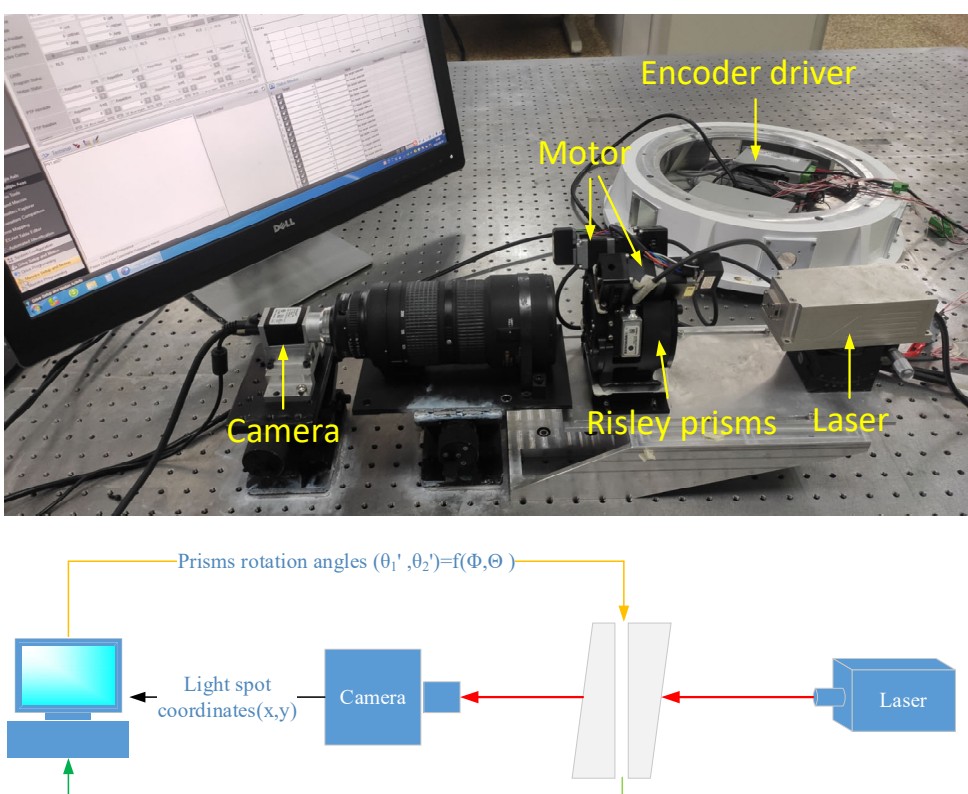

**Figure 11.** Experimental platform and schematic diagram.

We set $\Delta\theta = 1.62°$, $3.24°$, $4.86°$ and let the two prisms rotate for one circle in the same direction and speed, sampling at $1°$ intervals. A point every $10°$ was selected as the fitted data, for a total of 108 sets of data, and the remaining data were used as the validation set. Figure 12 shows the comparison of the pointing errors before and after error fit. Table 5 shows the maximum values, RMS values and descent rates of the forward solution error before and after error correction in the three experiments.

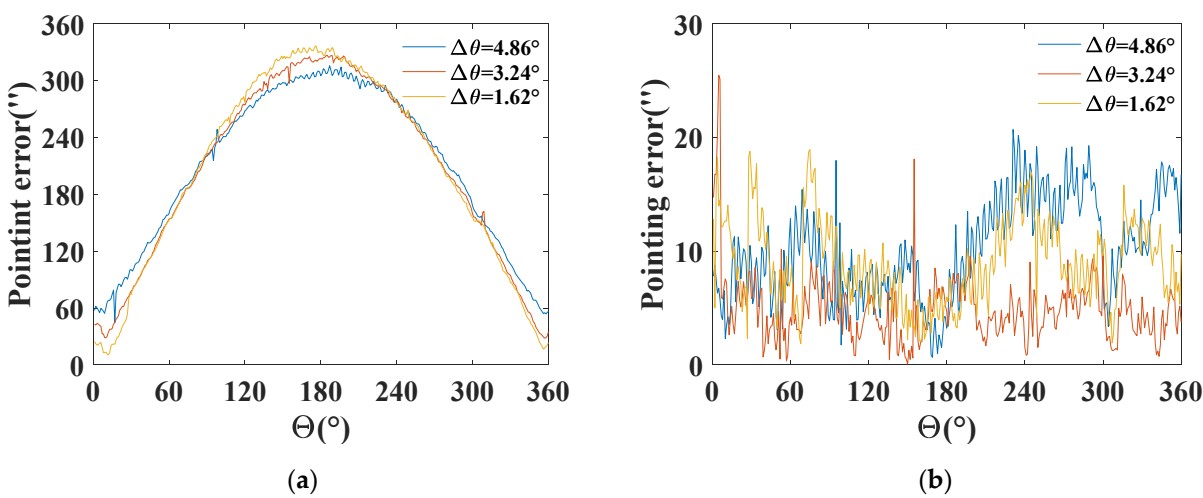

**Figure 12.** Comparison of the forward solution errors before and after correction in the experiment. (**a**) Before correction. (**b**) After correction.

**Table 5.** Experimental results of forward solution error correction.

| Deflection Angle | | Pointing Error | |
|---|---|---|---|
| | | **Maximum** | **RMS** |
| $\Delta\theta = 4.86°$ | Before correction | 336.6017″ | 224.4371″ |
| | After correction | 20.6913″ | 11.0432″ |
| | Descent rate | 93.76% | 95.08% |
| $\Delta\theta = 3.24°$ | Before correction | 326.7776″ | 226.0153″ |
| | After correction | 25.4662″ | 5.7213″ |
| | Descent rate | 92.21% | 97.47% |
| $\Delta\theta = 1.62°$ | Before correction | 336.6017″ | 229.6042″ |
| | After correction | 18.9351″ | 9.7448″ |
| | Descent rate | 94.37% | 95.76% |

We chose the deflection angle $\Phi = 0.1°$ and the azimuth angle $\Theta$ from $1°$ to $360°$ as the target pointing position to verify the inverse solution correction algorithm; the results are shown in Figure 13. Table 6 shows the maximum value, RMS value and descent rate of inverse solution error before and after error correction. The maximum pointing error is $0.0997°$ when using the two-step method directly for the inverse solution, and $0.0059°$ when using the correction method. The maximum error and the RMS value of errors decreased by 94.08% and 95.18%, respectively. The experiment proves the effectiveness of the reverse solution correction method.

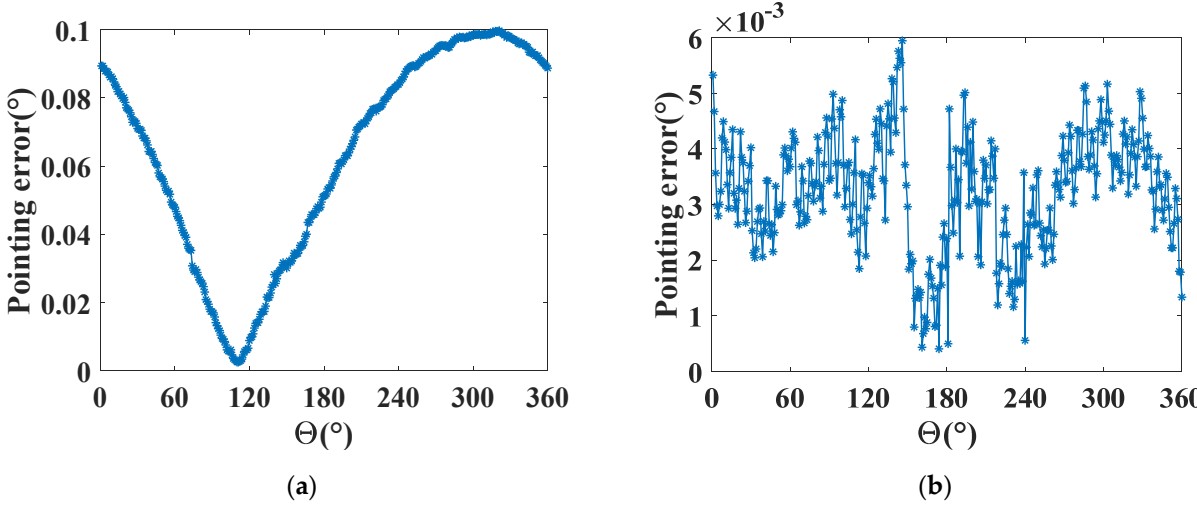

(a)

(b)

**Figure 13.** Comparison of the inverse solution error before and after correction in the experiment. (**a**) Before correction. (**b**) After correction.

**Table 6.** Experimental results of inverse solution error correction.

| | Pointing Error | |
|---|---|---|
| | **Maximum** | **RMS** |
| **Before correction** | 0.0997° | 0.0705° |
| **After correction** | 0.0059° | 0.0034° |
| **Descent rate** | 94.08% | 95.18% |

*4.3. Discussion*

With the same value of system error, the larger the maximum deflection angle of the Risley prisms, the larger the forward and inverse solution error caused by the system error. The maximum deflection angle of the Risley prisms in this article is $40°$, but it is difficult for experimental setups to measure large angles with high resolution. We can only

experimentally verify with the existing experimental setup in a small angle; simulated pointing experiments are performed by numerical simulation to verify the algorithm at large angles. The inverse solution error correction accuracy of the experimental results is higher than the simulation results because the residuals after error correction are larger in the case of a large deflection angle.

## 5. Conclusions

In this paper, the inverse solution errors of Risley prisms induced by incident light tilt and prism rotation axis tilt are investigated. A correction algorithm based on pointing-field transformation is proposed. Based on a more accurate error model of Risley prisms, the systemic errors are obtained by parameter identification, thereby obtaining an accurate forward solution. Since the actual beam trajectory and pointing field are deformed and deviated under the impact of incident light tilt and prism rotation axis tilt, pointing errors will be generated when using the two-step method for the inverse solution. We simulate the variation of the inverse solution error over the entire pointing field. A correction method based on pointing-field transformation is proposed to correct the inverse solution error. The simulation results show that the maximum value and RMS value of inverse solution error decrease by 80.9% and 87.16%, respectively. Since the experiment is limited to a small deflection angle range, the error descent rate of the experimental results is better than the simulation results. The maximum error and the RMS value of error in the experiment are reduced by 94.08% and 95.18%, respectively. The correction method provides valuable guidance for improving the accuracy of Risley prisms.

**Author Contributions:** Conceptualization, methodology, and theory, Y.S.; simulations, Y.S.; modelling and data analysis, Y.S.; experiments, Y.S.; writing—original draft, Y.S.; writing—review and editing, J.W., L.H. and S.W.; funding, supervision, and project administration, S.G. All authors have read and agreed to the published version of the manuscript.

**Funding:** This research received no external funding.

**Institutional Review Board Statement:** Not applicable.

**Informed Consent Statement:** Not applicable.

**Data Availability Statement:** Data supporting reported results can be obtained from the corresponding author.

**Conflicts of Interest:** The authors declare no conflict of interest.

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
