# Peer review of "Inverse Solution Error Analysis and Correction of Beam Steering System Based on Risley Prisms"

_applsci, doi:10.3390/app12041972_

Round 1
Reviewer 1 Report
Review of manuscript entitled “Inverse Solution Error Analysis and Correction of Beam Steering System Based on Risley Prisms” by Yinuo Song et al.
This manuscript describes in the firsts sections the developing of an accurate error model and the error generated by the inverse solution for the Risley prisms. Later in the manuscript an experiment is described.
The mathematical development is not commented in my review.
Regarding the experimental section, the following commentaries are shown.
Figure 11 shows a photograph and a diagram of the experimental set up. Section 4.2 mentions some characteristics of the prims bearings, encoders camera and camera lens, camera array of pixels and their size and the field of view.
With the suggestions that I mention next you could change or add information in the manuscript text. Besides you can describe the changes in the letter sent to me.
1 Regarding the laser it is not mentioned the brand nor the model. Besides its beam angle divergence and the wavelength should be mentioned. Also the distance from the laser to the camera should be mentioned.
2 The angle and diameter of the prisms are mentioned. But details about their surface flatness also should be shown. How this precision in the prisms flatness affects the spot size on the pixels arrays? and on the steering angle?.
3 In section 4.2 line 5 is mentioned a focal length. I suppose it is the focal length of the camera lens. How did you know the focal length with such a great precision at the fourth digit or 200 nanometers?. Mention in the text the tests or measurements you have done.
4 An important parameter that should be known is the size of the spot of light that reaches the array of sensors or pixels. This spot is due to the focusing of light by the camera lens. If it is big it will cover some amount of pixels. If it is small it will cover just some spots. This spot could be round or have an undetermined shape. Thus how can you be sure about the position of the spot on the pixels array. Not matter how well you can measure the prisms angle position with an encoder having a resolution of 0.002” the position of the spot at the pixels array should be known with precision.
5 How affects the laser beam divergence on the beam focused spot?. What is the ideal distance between the laser and the camera?
Reviewer 2 Report
Minor Revision
In this manuscript, the author presented an interested paper which investigate the inverse solution errors of Risley prisms and propose a correction method to improve the accuracy. The author analyze the impact of different parameters on the systemic errors and validate the error correction model with simulations and experiments.
Overall, this is a good manuscript. There are some issues remain to be addressed/clarified.
- Line 149, it is not clear what does the author mean “zero position”.
- What is the full name of “TLS”?
- Line 242, the author analyzes the impact of different parameters on the errors. However, the author needs to clarify these parameters are usually independent in operating this prism system.
- There are many parameters that have different level of influence on the error. It would be better if the author can make a schematic to show their level of influence and the way impact the accuracy in a clearer way.
- Why do Figure 10 (c) and (d) look the same?

Reviewer 3 Report
The authors present a solution to correct the inherent error of beam steering of Rysley prims due to mechanical and optical tolerance. The method is based on an error modeling which is fitted with a subset of true values. The article is well constructed and can be read without too much effort. The motivation behind the article is clearly explained in lines 63 to 85. However there is one main concern and some secondary remarks which need to be addressed:
Main concern:
The discussion is missing.
I acknowledged that the authors took care about the naming and the drawing for this not easy geometry. However there is a lack of consistency between the drawings.
A) In figure 2 the prisms are labeled I and II whereas in figure 3 they are labeled 1 and 2.
B) I presume that the vectors A are of length 1 and should not be mixed with the red vectors drawn in figure 2. I needed some time to understand that the value K, L and M are in fact the components of the last vector giving the direction of the beam to the target. The equation 2 is misleading and would be better within the text because it is a definition and not the result of a calculation.
C) The lower indices i and r are misleading when used with the upper indices 1, 2, 3 and 4. As such I did not fully understand the notation of Figure 3 neither the fact that considering (0,0,-1) as the incident ray labeled 1, the last ray could be labeled 5.
D) A vector named n and refractive index n1 and n2 are not a good choice of letters because they are not related.
Secondary remarks
- Line 96 (equation 1) A reference such as doi:10.1364/JOSAA.29.001356 would be welcome.
- Lines 119-121: The sentence seems wrong: what is a deflection angle in your sentence?
- Figure 5: (d) is missing in the caption.
- Figure 6: more explanation about the colors and so on should be added in the caption.
- In lines 69-70 and 74-75, the authors motivated their article by the fact the deviation angles were small. But in their experimental set-up, they acknowledged that the field of view of the objective used is small in regard to the possible beam deflections that could be made with the Rysley prims. This should be discussed in a missing "discussion" part.
- The distortion induced by the camera lens is not taken into account.
Small typing errors
Line 31: a space is missing before (ATP).
Line 85: method is misspelled.
Line 166: Why to change from vector "A" to "s" ?
Lines 175-176 & 274 & 333: Strange separation characters between the variables
Round 2
Reviewer 1 Report
The responses are clear.
Two misprints.
a) last line before section 2 heading. It says "S" it sould say "5".
b) Page 3, Section 2.1, first paragraph, third line before the last one. It says "Figure F". It should say "Figure 3".
Reviewer 3 Report
The authors gave a satisfactory response to my remarks. There is still no discussion chapter as suggested, but the added paragraph before the conclusion could possibly act as "small discussion."
However a true "discussion" part would add value to the paper.
